# Hierarchical Mamba Meets Hyperbolic Geometry: A New Paradigm for Structured Language Embeddings

## Abstract

Selective state-space models excel at long-sequence modeling, but their capacity for language representation – in complex hierarchical reasoning – remains under-explored. Most large language models rely on *flat* Euclidean embeddings, limiting their ability to capture latent hierarchies. To address this, we propose *Hierarchical Mamba (HiM)*, integrating efficient Mamba2 with hyperbolic geometry to learn hierarchy-aware language embeddings for deeper linguistic understanding. Mamba2-processed sequences are projected to the Poincaré ball or Lorentzian manifold with "learnable" curvature, optimized with a hyperbolic loss. Our HiM model facilitates the capture of relational distances across varying hierarchical levels, enabling effective long-range reasoning for tasks like mixed-hop prediction and multi-hop inference in hierarchical classification. Experimental results show both HiM effectively capture hierarchical relationships across four linguistic and medical datasets, surpassing Euclidean baselines, with HiM-Poincaré providing fine-grained distinctions with higher h-norms, while HiM-Lorentz offers more stable, compact, and hierarchy-preserving embeddings-favoring robustness[1]

## 1 Introduction

Large language models (LLMs), such as Transformers (Vaswani, 2017) and BERT (Devlin et al., 2019), typically encode input sequences into a *flat* Euclidean space. However, they struggle to capture the hierarchical and tree-like structures inherent in natural language (Chomsky, 2014), often leading to distortions at different levels of abstraction and specificity (Nickel & Kiela, 2017; Ganea et al., 2018). Moreover, transformer-based encoders face significant computational overhead due to the quadratic complexity of the attention mechanism (Vaswani, 2017). This limitation becomes particularly evident when dealing with hierarchical data (e.g., text ontologies, brain connectome (Ramirez et al., 2025; Baker et al., 2024)) with exponentially expanding structure. State-space models, starting with the Structured State Space (S4) model (Gu et al., 2021), have shown exceptional scalability for long-sequence modeling. Mamba's selective mechanism (Gu & Dao, 2023) dynamically prioritizes relevant information, achieving state-of-the-art performance in tasks with long-range dependencies. Mamba2 refines the original Mamba model for long-range sequence tasks by introducing a duality between state-space computations and attention-like operations, enabling the model to function as either an SSM or a structured, "mask-free" form of attention (Dao & Gu, 2024).

Recently, leveraging hyperbolic geometry as the latent representation space in machine learning models has shown great promise for learning meaningful hierarchical structures (Nickel & Kiela, 2018; Peng et al., 2021; Petrovski, 2024). The Poincaré disk and Lorentz model are two prevalent representations of hyperbolic space. The Poincaré disk model is often favored for its conceptual simplicity (bounded in a unit ball). However, the Lorentz model (with unbounded infinite space) offers a closed-form distance function, but requires careful handling of numerical functions dealing with space-like dimensions and a time-like dimension using exponential mapping and logarithmic mapping (Peng et al., 2021). These numerical considerations are critical because the improper handling of the time-like coordinate in Lorentz models can lead to manifold violations, requiring specialized projection techniques (Fan et al., 2024; Liang et al., 2024). Existing hyperbolic LLM architectures (He et al., 2024b; Peng et al., 2021) often rely on Transformer blocks and apply a simple

---

[1]The source code is publicly available at `https://github.com/BerryByte/HiM`.

Poincaré disk model, leading to $O(L^2)$ complexity that becomes prohibitive for long sequences typical in deep hierarchies. A key challenge in implementing hyperbolic models is tuning the curvature to maintain numerical stability, particularly in Lorentz parameterization.

In this paper, we introduce hyperbolic mamba with the Lorentz model, and compare it with its counterparts – *Poincaré model and Euclidean model*. To address the potential numerical instability in the Lorentz model (Mishne et al., 2023), we explicitly bound the embedding norms and employ curvature-constrained Maclaurin approximations for hyperbolic operations. HiM aims to achieve high-performance hierarchical classification by preserving relational hierarchies. It demonstrates scalability for processing long sequences without compromising on accuracy or computational efficiency. HiM's novelty lies in integrating a state-space model (Mamba2) with hyperbolic geometry, leveraging Mamba2's $O(L)$ complexity for efficient sequence modeling while preserving hyperbolic properties for hierarchical representation. Additionally, our HiM incorporates novel task-specific hyperbolic losses that explicitly enforce parent-child distance constraints in hyperbolic space, enabling end-to-end hierarchy learning without Euclidean biases and achieving significant F1 gains on multi-hop inference tasks. To support HiM's framework, we introduce **SentenceMamba-16M**, a compact, Mamba2-based large language model with 16 million parameters designed to generate high-quality sentence embeddings.

## 2 RELATED WORKS

Hyperbolic geometry has demonstrated strong potential in modeling hierarchical structures in both shallow and deep neural networks. Foundational works, such as Poincaré embeddings (Nickel & Kiela, 2017) and hyperbolic entailment cones (Ganea et al., 2018), showed their effectiveness in capturing hierarchical relationships in taxonomies with shallow neural networks. Moreover, hyperbolic manifolds have also been applied to encode hierarchies in graph-structured data (Liu et al., 2019; Chami et al., 2019). More recent efforts have extended hyperbolic representations to multimodal computer vision tasks, including visual and audio modalities (Yang et al., 2024c; Mandica et al., 2024), further demonstrating their strength in capturing both hierarchical structure and uncertainty.

However, hyperbolic approaches in language modeling remain limited. As an early approach to hyperbolic word embeddings, Dhingra et al. (2018) provided an important early step by reparameterizing Poincaré embeddings for GRU-based sequence modeling. This approach eliminated projection steps and supported both shallow and parametric encoders, but it was ultimately limited by its shallow representations, which restricted its expressive power and ability to capture long-range dependencies. More recent research has extended these concepts to transformers and their variants (He et al., 2024b; Chen et al., 2021; 2024). These approaches enable effective prediction of subsumption relations and transitive inferences across hierarchy levels using hyperbolic embeddings, providing a principled framework for encoding syntactic dependencies through geodesic distances. However, Hyperbolic BERT exhibits high computational cost than standard BERT due to the complexity of hyperbolic operations (Chen et al., 2024). To improve efficiency, recent works have explored finetuning LLMs directly in hyperbolic space with the Low-Rank Adaptation (LoRA) technique (Hu et al., 2022). For example, HoRA (Yang et al., 2024a) and HypLoRA (Yang et al., 2024b) apply LoRA to the hyperbolic manifold, allowing parameter-efficient fine-tuning while capturing complex hierarchies. These methods show strong gains—up to 17.3% over Euclidean LoRA. However, these models usually assume a constant curvature, which may not be optimal for all data, and can suffer from numerical instability due to the exponential and logarithmic mappings required to transition between Euclidean and hyperbolic spaces (López & Strube, 2020).

**Limitations in Current Approaches and Our Contribution:** Despite significant progress, most existing methods either exploit only partial hyperbolic representations (e.g., using adapters or static embeddings) or rely heavily on attention-based architectures that scale poorly with long sequences and deep hierarchies. For instance, Poincaré GloVe (Tifrea et al., 2018) is limited to word embeddings, failing to capture dynamic, context-dependent relationships, while Hyperbolic BERT (Chen et al., 2024) and HiT (He et al., 2024b) introduce significant computational overhead, especially for long sequences. Similarly, probing BERT's embeddings in a Poincaré ball (Chen et al., 2021) to analyze hierarchical structures, but their diagnostic approach does not train a new model for hierarchical reasoning tasks like HiM. Methods, such as HoRA (Yang et al., 2024a) and HypLoRA (Yang et al., 2024b), only introduce hyperbolic geometry through adapter modules added post hoc to standard transformer backbones. These methods inherit the architectural inefficiencies of transformers but

cannot fully encode hierarchy directly within the hyperbolic latent space. Building on the strengths and limitations discussed above, we propose *Hierarchical Mamba (HiM)* as a novel framework for long-range hierarchical reasoning. Our contributions can be summarized as follows:

- **Direct hyperbolic integration:** Unlike prior works using adapters or pre/post-processing, HiM projects sentence-level Mamba2 representations directly into hyperbolic manifolds (Poincaré and Lorentzian), embedding hierarchy at the core of the model's design.
- **SentenceMamba-16M:** We introduce *SentenceMamba-16M*, a Mamba2-based LLM (16M parameters) trained at sentence-level embeddings on the SNLI dataset (Bowman et al., 2015).
- **Stabilized hyperbolic operations:** HiM addresses numerical instability in Lorentzian manifolds using curvature-bounded Maclaurin approximations for hyperbolic functions, ensuring robust training for deep hierarchies.
- **Novel hyperbolic losses:** HiM employs weighted clustering and centripetal losses to enforce parent-child separation and compact clustering of related entities with respect to origin, enhancing hierarchical structure preservation in hyperbolic space.

## 3 METHODOLOGY

Hyperbolic geometry, characterized by negative curvature $\mathcal{K} = -1/c$, is well-suited for hierarchical data due to its exponential growth properties, modeled using the Poincaré ball or Lorentz model (Nickel & Kiela, 2017; 2018). Mamba2, a state-space model (SSM), offers efficient sequence modeling with linear complexity, using structured state-space duality to balance SSM and attention-like operations (Dao & Gu, 2024). The detailed formulations and preliminaries of Mamba2 are provided in Appendix A.1.

### 3.1 HYPERBOLIC MAMBA (HIM)

The overall framework of HiM, including the integration of Mamba2 blocks and hyperbolic projections, is shown in Figure 1. Firstly, the raw text is tokenized into a sequence of tokens; these token IDs are mapped into embedded tokens resulting in token IDs of shape $[B, L, D, N]$, where $B$ is the batch size ($B = 256$), $L$ is the sequence length ($L = 128$), and $D$ is the embedding dimension ($D = 384$), and N is the state dimension ($N = 96$). These embeddings are passed through a sequence of 4 Mamba2 Blocks. Alanis-Lobato et al. (2016) focuses on efficient embedding of complex networks into hyperbolic space using the network Laplacian, achieving a computational complexity of $O(N^2)$ and enabling the analysis of large networks in seconds. This highlights the importance of computational efficiency in scaling hyperbolic models, a principle that HiM extends by Mamba2 blocks (Equations 17 and 20) to achieve linear-time complexity $O(L)$, making it particularly suited for long sequences and deep hierarchical language structures. In the Mamba2 blocks, the inputs $B$ having 384 dimensions are projected into the intermediate state $I$ having 768 dimensions using a linear transformation ($D : 384 \rightarrow I : 768$).

The $x'_t$ component undergoes a convolution operation with a kernel size of 4. A SiLU activation function follows this operation for non-linearity. A detailed implementation of the Mamba2 block is discussed in Appendix A.1. In the output projection, the intermediate state dimensions are projected back to their original embedding dimension $I : 768 \rightarrow D : 384$ for compatibility with downstream tasks. The SentenceMamba-16M model, central to HiM, is randomly initialized with Kaiming normal weights instead of pretrained weights, enabling it to learn hierarchical structures directly from training data in hyperbolic space without any biases. The SentenceMamba-16M model is trained using Triplet Contrastive Loss, which brings embeddings of positive pairs closer together while pushing apart embeddings of other sentences in the batch. This loss function has proven effective in prior works involving hierarchical embedding (Schroff et al., 2015; He et al., 2024b). After normalizing each embedding to unit length, we measure the pairwise cosine similarity as $\text{sim}(i, j) = \mathbf{e}_i \cdot \mathbf{e}_j$, where each embedding $\mathbf{e}_i$ and $\mathbf{e}_j$ belong to $\mathbf{e} \in \{\mathbf{e}_1, \mathbf{e}_2, \ldots, \mathbf{e}_n\}$. We then calculate the contrastive loss for the batch by constructing a similarity matrix from the similarity scores across nodes. The sentence embeddings are constrained using hyperbolic tangent activation followed by L2 normalization to ensure numerical stability:

$$\mathbf{u} = \text{normalize}(\tanh(s)), \tag{1}$$

where $\mathbf{s}$ is the mean-pooled embedding from the Mamba2 blocks and normalize($\cdot$) denotes L2 normalization. This operation reduces the sequence to a single fixed-size vector, representing the pooled features of the entire input sequence normalized to unit sphere.

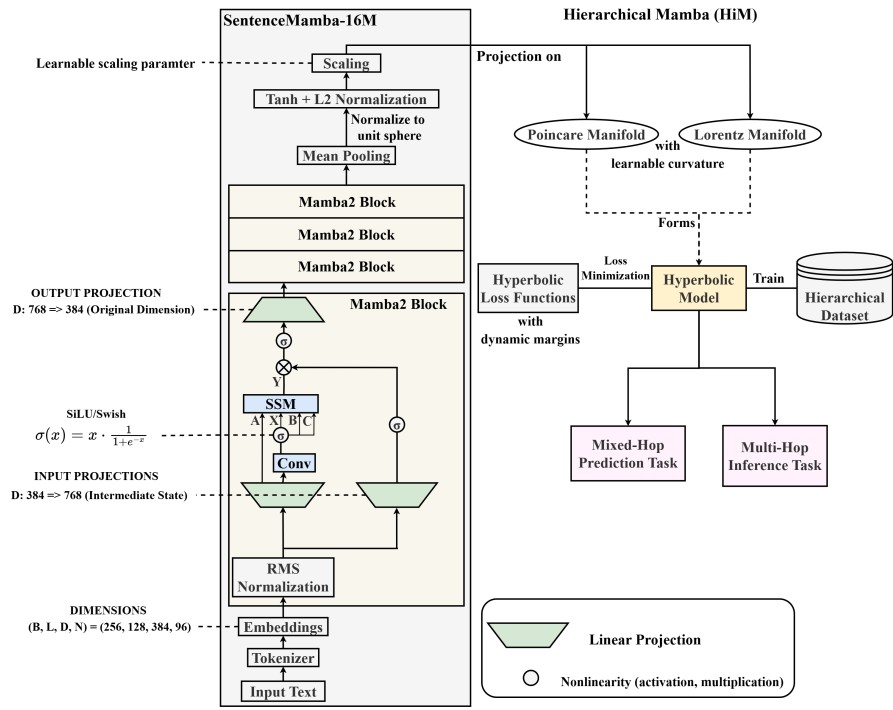

Figure 1: Overview of the Hierarchical Mamba (HiM) model, integrating Mamba2 blocks with hyperbolic projections to the Poincaré ball (via tangent-based mapping) and Lorentzian manifold (via cosine/sine-based mapping), enabling efficient and hierarchy-aware language embeddings for long-range reasoning tasks.

To ensure numerical stability during hyperbolic projection, we apply norm scaling with a learnable parameter $\gamma$:

$$\mathbf{h} = \begin{cases} \gamma \cdot \mathbf{u}, & \text{for Poincaré,} \\ \gamma \cdot \text{clamp}(\mathbf{u}, -8, 8), & \text{for Lorentz.} \end{cases} \tag{2}$$

This approach mitigates numerical overflow and enhances training stability. The interplay between the norm scaling parameter $\gamma$ and curvature $\mathcal{K} = -1/c$ is mathematically significant. When we scale embeddings by $\gamma$ before projection, it effectively modulates the curvature of the hyperbolic space. For the curvature parameter $c$, scaling the norm of $\mathbf{h}$ by $\gamma$ leads to an effective curvature of $\mathcal{K}_{\text{eff}} = -1/(c\gamma^2)$. This can be seen from the Poincaré ball projection Equation 3, where scaling $\|\mathbf{h}\|$ by $\gamma$ is equivalent to modifying $\sqrt{c}$ by a factor of $1/\gamma$. Thus, the model learns both a base curvature $c$ and a scaling factor $\gamma$ that together determine the optimal geometry for representing hierarchical relationships. This dual learning approach provides flexibility in adapting the hyperbolic space to the structural complexity of the data while maintaining numerical stability. Then the vector $h$ is mapped to a point $e$ in hyperbolic space. The general form for a Poincaré ball with radius $r = \sqrt{c}$ (curvature $\mathcal{K} = -1/c = -1/r^2$) is:

$$e_P = \sqrt{c} \cdot \tanh\left(\frac{\|\mathbf{h}\|}{\sqrt{c}}\right) \cdot \frac{\mathbf{h}}{\|\mathbf{h}\|}, \tag{3}$$

where $\|h\|$ is the norm of $h$. This scaling ensures the vector lies within the unit ball. This yields the final sentence embedding $e$ with values constrained between $-1$ and $1$ (indicating a positive or negative parent), allowing us to project the embedding onto the Poincaré Ball manifold.

We also project it onto the Lorentzian manifold as it yields richer features in a more convenient original hyperbolic space. The pooled embeddings are instead mapped to the Lorentzian manifold using:

$$e_L = \begin{bmatrix} \sqrt{c} \cdot \cosh\left(\frac{\|\mathbf{h}\|}{\sqrt{c}}\right) \\ \sqrt{c} \cdot \sinh\left(\frac{\|\mathbf{h}\|}{\sqrt{c}}\right) \cdot \frac{\mathbf{h}}{\|\mathbf{h}\|} \end{bmatrix}. \tag{4}$$

Here, $h$ is the norm distance, $\sqrt{c}$ is the radius of the hyperbolic space; $\mathcal{K} < 0$ ensures hyperbolic geometry. For the Lorentz mapping we let $z = \|\mathbf{h}\|/\sqrt{c}$ in Equation 4. The **cosh, sinh** functions are the Hyperbolic cosine and sine functions used to compute projections. The first term $\sqrt{c} \cdot \cosh\left(\frac{\|\mathbf{h}\|}{\sqrt{c}}\right)$ in the Lorentz projection is the time-like dimension. The remaining components $\sqrt{c} \cdot \sinh\left(\frac{\|\mathbf{h}\|}{\sqrt{c}}\right) \cdot \frac{\mathbf{h}}{\|\mathbf{h}\|}$ are the space-like dimension. This step is crucial for hyperbolic geometry as it ensures the embeddings are bounded, enabling seamless projection onto the Lorentzian manifold.

While the Lorentz projection typically uses exact hyperbolic functions, we stabilize the training even more by approximating $\cosh$ and $\sinh$ via their Maclaurin (Taylor) expansions for $|z| < 10^{-3}$. By substituting truncated polynomial expansions, we limit overflow and hence solve the problem of exploding gradients.

$$\cosh(z) = 1 + \frac{z^2}{2!} + \frac{z^4}{4!} + \cdots; \qquad \sinh(z) = z + \frac{z^3}{3!} + \frac{z^5}{5!} + \cdots. \tag{5}$$

Then, the first (time-like) coordinate and the remaining (space-like) coordinates from Equation 4 become:

$$\mathbf{e_L} \approx \begin{bmatrix} \sqrt{c}\left(1 + \frac{z^2}{2} + \frac{z^4}{24} + \cdots\right) \\ \sqrt{c}\left(z + \frac{z^3}{6} + \frac{z^5}{120} + \cdots\right)\frac{\mathbf{h}}{\|\mathbf{h}\|} \end{bmatrix}. \tag{6}$$

To fully exploit the hyperbolic structure of our model, we employ an advanced hyperbolic loss function for the HiM model optimization, which is a weighted combination of centripetal loss and clustering loss. These losses enhance the model's ability to effectively learn hierarchical relationships by optimally positioning and grouping the embeddings in a strongly hierarchical structure within the Hyperbolic manifold. Detailed equations for our hyperbolic loss are presented below, with the full calculation process provided in Appendix A.2.

**Centripetal Loss:** This loss function ensures that parent entities are positioned closer to the origin of the hyperbolic manifold than their child counterparts. This reflects the natural expansion of hierarchies from the origin to the boundary of the manifold.

$$\mathcal{L}_{\text{centri}} = \sum_{(e,e^+,e^-)\in\mathcal{D}} \max(\|e^+\|_c - \|e\|_c + \beta, 0) \quad \text{where} \quad \|\cdot\|_c := d_c(\cdot, 0). \tag{7}$$

**Clustering Loss:** This loss function clusters related entities and distances unrelated ones within the hyperbolic manifold, promoting the grouping of similar entities while preserving hierarchical separation.

$$\mathcal{L}_{\text{cluster}} = \sum_{(e,e^+,e^-)\in\mathcal{D}} \max(d_c(e, e^+) - d_c(e, e^-) + \alpha, 0). \tag{8}$$

Here, $(e, e^+, e^-)$ represent the hyperbolic embeddings of a randomly selected anchor node, its positive parent node, and an unrelated negative node, respectively. $\|e\|_c$ or $d_c(e, 0)$ measures the distance from the origin to the hyperbolic embedding $e$ in the Poincaré and Lorentzian manifold. $d_c(e, e^+)$ measures the distance between hyperbolic embeddings of node $e$ and its positive parent node $e^+$. $d_c(e, e^-)$ measures the distance between node $e$ and a negative node $e^-$. $\alpha$ and $\beta$ denote margin parameters to enforce centripetal and clustering properties, respectively.

The Hyperbolic Loss $\mathcal{L}_{hyperbolic}$ is defined as the weighted sum of Centripetal Loss $\mathcal{L}_{centri}$ and Clustering Loss $\mathcal{L}_{cluster}$:

$$\mathcal{L}_{hyperbolic} = w_{ce}\mathcal{L}_{centri} + w_{cl}\mathcal{L}_{cluster}, \tag{9}$$

where $w_{ce}$ and $w_{cl}$ are weights that control the contribution of each loss component. This loss ensures that the model maintains the hierarchical structure during training, with parent entities closer to the origin and related entities clustered together. The margins $\alpha$ and $\beta$ in the clustering and centripetal losses (Equations 7 and 8) are implemented as dynamic parameters optimized during training to adaptively enforce the hierarchical constraints. The margins for the clustering and centripetal losses are adapted to the hyperbolic geometry by scaling proportionally with the radius $r = \sqrt{c}$, ensuring the loss functions remain geometrically consistent across different curvatures. These scaling

factors were determined through empirical validation to maintain consistent separation properties as the model adapts its curvature during training. The clustering margin is intentionally larger to enforce robust hierarchical separation between related and unrelated entities, while the centripetal margin is smaller to allow fine-grained positioning of parent nodes closer to the origin relative to their children, reflecting the natural expansion of hierarchies in hyperbolic space. In all hierarchical classification tasks, hard negatives were chosen to sharpen the model's discrimination (Schroff et al., 2015). Rather than randomly sampling unrelated nodes, we select negative examples that are semantically close to the anchor (or positive) in embedding space. This training strategy forces the model to learn more subtle hierarchical distinctions, which is crucial for tasks such as "multi-hop inference". We observe that hard negatives lead to better generalization.

Following Chami et al. (2019), we optimize the learnable curvature parameter using the AdamW optimizer. This is justified because the curvature parameter itself is a scalar Euclidean variable controlling the hyperbolic manifold geometry, making AdamW both theoretically valid and empirically stable. To ensure numerical stability during training in hyperbolic space as the curvature adapts, we implement a geometric stabilization technique that periodically projects the model parameters back onto the manifold. Specifically, every 100 optimization steps, this stabilization counteracts numerical drift that can occur during curvature optimization, preventing embeddings from violating the constraints of the hyperbolic geometry and ensuring all distance computations remain well-defined throughout training.

## 4 EXPERIMENTS

**Dataset** We compare our proposed HiM models with their Euclidean counterparts, evaluated across four ontology datasets (i.e., DOID, FoodOn, WordNet, and SNOMED-CT) varying in scale and hierarchical complexity.[2]. (1) DOID offers a structured representation of human diseases through "is-a" relationships (Schriml et al., 2012). (2) FoodOn is a detailed ontology that standardizes food-related terminology, covering ingredients, dishes, and processes for nutritional classification and dietary research. It uses a hierarchical structure and borrows from existing ontologies like LanguaL (Dooley et al., 2018). (3) WordNet is a well-known benchmark that organizes English nouns, verbs, and adjectives into synonym sets connected by hypernym-hyponym relationships (Miller, 1995). (4) SNOMED Clinical Terms (SNOMED-CT) is a comprehensive clinical terminology system used in electronic health records (EHRs). It organizes concepts (e.g., diagnoses, procedures, symptoms) into multiple hierarchies, linked by "is-a" and attribute relationships (Stearns et al., 2001). All datasets are derived from structured taxonomies and can be represented as directed acyclic graphs, where nodes denote entities and edges denote direct subsumption (i.e., parent-child) relations.

**Implementation Details** We use 4 NVIDIA A100 GPUs with 80GB of memory each, distributed across a single compute node. Our model is implemented using the mamba-ssm library (Dao & Gu, 2024). To define and operate over hyperbolic manifolds, we use GeoOpt (Kochurov et al., 2020), while DeepOnto (He et al., 2024a) is employed to process and manage hierarchical structures in the ontology datasets. We leverage distributed data-parallel training with PyTorch's DistributedDataParallel wrapper (Paszke, 2019). Our models were trained for ten epochs using the AdamW optimizer with a linear warm-up learning rate over the first 100 steps (target learning rate set to $1e-4$), and weight decay of $1e-3$. The linear warm-up is followed by a constant learning rate $1e-4$. The maximum gradient norm is clipped to 1.0. We employ a combination of hyperbolic clustering loss and hyperbolic centripetal loss during pretraining, with weights of 1.0 and 1.0, respectively. Our model incorporates several learnable parameters, such as scaling factor $\gamma$ (initialized to 0.01), curvature $c$ (initialized to 1). We implement dynamic margin parameters for losses $\alpha$ and $\beta$, which depend on the updated curvature. We use a batch size of 256 per GPU. To regularize the model during training, a dropout rate of 0.2 is applied following each Mamba2 block. The detailed train/validation/test splits for mixed-hop prediction and multi-hop inference tasks, can be found from Table 3 in Appendix A.3.

**Evaluation Tasks and Metrics** We evaluated our HiM models on two key tasks designed to assess its hierarchical reasoning capabilities in ontology completion and knowledge graph inference: (1) *multi-hop inference*, which involves predicting the existence of indirect relationships (e.g., "dog is

---

[2]Datasets are available from `https://zenodo.org/records/14036213` and see Table 3 in Appendix A.3 for details

a vertebrate") through transitive reasoning. (2) *mixed-hop prediction*, which focuses on estimating hierarchical distances between entities (e.g., 1-hop vs. 2-hop relations). Both tasks are formulated as classification problems based on hyperbolic distances. Detailed formulations are provided in Appendix A.4. We use three metrics for evaluation: F1 score, Precision, and Recall. Among them, the F1 score serves as the primary metric, as it provides a balanced measure of precision and recall, which is critical for hierarchical reasoning tasks. Following prior work on these datasets (He et al., 2024b), we exclude Accuracy due to its vulnerability to class imbalance, where negative samples significantly outnumber positive ones. During training, models are optimized using entity triplets (anchor, positive, negative) under a contrastive learning framework; however, evaluation is performed on entity pairs to directly assess subsumption prediction performance.

## 5   RESULTS

We compare our proposed HiM models–HiM-Poincaré and HiM-Lorentz– against two Euclidean baselines (pretrained SentenceMamba-16M and finetuned SentenceMamba-16M) on four hierarchical datasets for two main downstream tasks: mixed-hop prediction and multi-hop inference. The pretrained SentenceMamba-16M is obtained by training on the SNLI dataset (Bowman et al., 2015), while the fine-tuned SentenceMamba-16M model is initialized using Kaiming normal initialization for weights and zero initialization for biases, then fine-tuned on hierarchical datasets. Our HiM models share the SentenceMamba-16M backbone ($\approx$ 16M parameters), but incorporate learnable curvature and are trained in hyperbolic space using both Poincaré and Lorentzian manifolds.

### 5.1   COMPARISON BETWEEN HIM MODELS AND THEIR EUCLIDEAN BASELINES

We present a comprehensive comparison of HiM-Poincaré, HiM-Lorentz, and their Euclidean baselines, including the pretrained SentenceMamba-16M and the fine-tuned SentenceMamba-16M (randomly initialized and trained on hierarchical datasets in *Euclidean space*), in Table 1. Both HiM models were trained with learnable curvature parameter $\mathcal{K} = -1/c$. A deeper curvature (smaller radius $r =¿$ smaller $c =¿$ larger/deeper curvature $\mathcal{K}$) allows us to exploit the hierarchical structure of the Hyperbolic manifold much better, as the hyperbolic embeddings are confined in the conical manifold compact within a smaller radius. The average $\delta$-hyperbolicity (Gromov, 1987) for each dataset measures the tree-likeness of the graph by calculating the maximum deviation from the four-point condition. Values closer to 0 indicate a more hierarchical structure (Adcock et al., 2013), making these datasets well-suited for hyperbolic embeddings. The corresponding $\delta$-hyperbolicity scores for the four datasets are reported in Table 1, reflecting a descending order of hierarchy complexity: DOID $\rightarrow$ SNOMED-CT $\rightarrow$ WordNet $\rightarrow$ FoodOn. The experimental results illustrate that HiM-Lorentz model achieves more robust and stable performance (with extremely small variance) in terms of the F1, precision, and recall values for both mixed-hop prediction and multi-hop inference tasks across four datasets. Moreover, HiM-Lorentz outperformed the HiM-Poincaré variant on the multi-hop inference task for both the WordNet and SNOMED-CT datasets, both are relatively large datasets and exhibit deeper hierarchies characterized by small $\delta$-hyperbolicity. However, in the case of FoodOn—which also has *higher hyperbolicity*—the Poincaré-based model achieved better performance.

### 5.2   COMPARISON WITH HYPERBOLIC TRANSFORMER BASELINE

To further evaluate the effectiveness of HiM, we compare it with a hyperbolic transformer model (HiT) (He et al., 2024b), which was designed by fine tuning a pretrained model (i.e., all-MiniLM-L6-v2) with low curvature ($\mathcal{K} = -1/384$, nearly Euclidean). To ensure a fair comparison while preserving the hyperbolic nature of the space, we randomly initialize our HiM and evaluate it against a randomly initialized version of HiT (HiT*). We conducted ablations across two different curvatures ($\mathcal{K} = -1.0$ and $-1/d$ where embedding dimension $d = 384$) and model sizes (16M and 32M parameters) on both Poincaré and Lorentz manifolds. The results in Table 2 show that HiM consistently outperforms the transformer-based HiT* model across both manifolds and most experimental configurations.

Notably, performance degrades significantly under low curvature settings ($\mathcal{K} = -1/d$), particularly in the Lorentz manifold, where HiT* shows substantial performance drops. This suggests that stronger hyperbolic curvature ($\mathcal{K} = -1.0$) is essential for effective hierarchical modeling. Under our fixed curvature, Poincaré's bounded nature enables more stable norm dispersion and discriminative gradient flow, particularly when combined with variance regularization in our centripetal loss. In contrast, Lorentz embeddings tend to collapse toward the hyperboloid's shell where time-like distances flatten and norm-based separation weakens. To provide broader context for our results, we

Table 1: Performance comparison of Pretrained, Finetuned across Euclidean manifold, and HiM models across Hyperbolic manifolds on various datasets (with varying average $\delta$-hyperbolicity). Pretrained SentenceMamba-16M is trained on SNLI; Finetuned and HiM load the pretrained model, but randomly initialize it before training. HiM uses learnable curvature for hyperbolic projections. The mean and standard deviation of F1, Precision and Recall scores were computed over five independent runs for each setting. (See details in Appendix A.6)

| Metric | Euclidean ($\mathcal{K} = 0$) | | Hyperbolic ($\mathcal{K} < 0$, learnable) | |
|---|---|---|---|---|
| | Pretrained | Finetuned | HiM-Poincaré | HiM-Lorentz |
| **Mixed-hop Prediction (DOID) : Average $\delta$-Hyperbolicity = 0.0190** | | | | |
| F1 | $0.135 \pm 0.022$ | $0.436 \pm 0.043$ | $0.795 \pm 0.019$ | $\mathbf{0.821 \pm 0.003}$ |
| Precision | $0.087 \pm 0.003$ | $0.776 \pm 0.016$ | $0.812 \pm 0.020$ | $\mathbf{0.822 \pm 0.004}$ |
| Recall | $0.390 \pm 0.207$ | $0.305 \pm 0.040$ | $0.780 \pm 0.026$ | $\mathbf{0.820 \pm 0.007}$ |
| **Mixed-hop Prediction (FoodOn) : Average $\delta$-Hyperbolicity = 0.1852** | | | | |
| F1 | $0.125 \pm 0.046$ | $0.550 \pm 0.017$ | $\mathbf{0.836 \pm 0.031}$ | $0.827 \pm 0.002$ |
| Precision | $0.090 \pm 0.009$ | $0.688 \pm 0.008$ | $0.841 \pm 0.024$ | $\mathbf{0.852 \pm 0.007}$ |
| Recall | $0.330 \pm 0.232$ | $0.459 \pm 0.023$ | $\mathbf{0.831 \pm 0.033}$ | $0.803 \pm 0.002$ |
| **Mixed-hop Prediction (WordNet) : Average $\delta$-Hyperbolicity = 0.1438** | | | | |
| F1 | $0.135 \pm 0.044$ | $0.615 \pm 0.009$ | $\mathbf{0.824 \pm 0.024}$ | $0.823 \pm 0.003$ |
| Precision | $0.086 \pm 0.014$ | $0.755 \pm 0.018$ | $\mathbf{0.853 \pm 0.023}$ | $0.828 \pm 0.006$ |
| Recall | $0.430 \pm 0.238$ | $0.519 \pm 0.006$ | $0.798 \pm 0.029$ | $\mathbf{0.815 \pm 0.004}$ |
| **Mixed-hop Prediction (SNOMED-CT) : Average $\delta$-Hyperbolicity = 0.0255** | | | | |
| F1 | $0.129 \pm 0.017$ | $0.672 \pm 0.009$ | $0.886 \pm 0.027$ | $\mathbf{0.890 \pm 0.004}$ |
| Precision | $0.084 \pm 0.001$ | $0.886 \pm 0.003$ | $0.894 \pm 0.024$ | $\mathbf{0.901 \pm 0.006}$ |
| Recall | $0.375 \pm 0.207$ | $0.541 \pm 0.012$ | $0.877 \pm 0.032$ | $\mathbf{0.880 \pm 0.005}$ |
| **Multi-hop Inference (WordNet) : Average $\delta$-Hyperbolicity = 0.1431** | | | | |
| F1 | $0.134 \pm 0.045$ | $0.648 \pm 0.012$ | $0.865 \pm 0.026$ | $\mathbf{0.872 \pm 0.004}$ |
| Precision | $0.086 \pm 0.016$ | $0.768 \pm 0.012$ | $0.867 \pm 0.023$ | $\mathbf{0.871 \pm 0.007}$ |
| Recall | $0.431 \pm 0.240$ | $0.560 \pm 0.013$ | $0.863 \pm 0.031$ | $\mathbf{0.872 \pm 0.005}$ |
| **Multi-hop Inference (SNOMED-CT) : Average $\delta$-Hyperbolicity = 0.0254** | | | | |
| F1 | $0.128 \pm 0.016$ | $0.630 \pm 0.010$ | $0.919 \pm 0.028$ | $\mathbf{0.920 \pm 0.003}$ |
| Precision | $0.083 \pm 0.001$ | $0.902 \pm 0.002$ | $0.917 \pm 0.024$ | $\mathbf{0.919 \pm 0.008}$ |
| Recall | $0.369 \pm 0.205$ | $0.483 \pm 0.011$ | $\mathbf{0.921 \pm 0.034}$ | $0.920 \pm 0.008$ |

Table 2: F1 scores comparing HiM models with HiT* across different parameter scales and curvature settings on mixed-hop prediction tasks. HiT* (Hyperbolic Transformer with random initialization) uses identical hyperbolic projections, loss functions, and manifolds as HiM, differing only in the use of Transformer architecture instead of Mamba. Results demonstrate consistent advantages of the Mamba architecture over Transformers in hyperbolic settings.

| #Param. | Curvature ($\mathcal{K}$) | HiM-Poincaré | | HiT*-Poincaré | | HiM-Lorentz | | HiT*-Lorentz | |
|---|---|---|---|---|---|---|---|---|---|
| | | WordNet | DOID | WordNet | DOID | WordNet | DOID | WordNet | DOID |
| 16M | $-1.0$ | **0.859** | **0.902** | 0.846 | 0.837 | **0.850** | **0.890** | 0.838 | 0.844 |
| 16M | $-1/d$ | 0.788 | **0.863** | **0.840** | 0.825 | **0.781** | **0.704** | 0.525 | 0.520 |
| 32M | $-1.0$ | **0.849** | **0.883** | 0.836 | 0.831 | **0.837** | **0.869** | 0.829 | 0.832 |
| 32M | $-1/d$ | 0.769 | **0.839** | **0.809** | 0.835 | **0.815** | **0.711** | 0.465 | 0.530 |

include additional analysis with comparing our model to GPT-4o on zero-shot mixed-hop prediction and also study HiM's computational efficiency in terms of sequence length in Appendix A.7.

### 5.3 VISUALIZATION OF HYPERBOLIC EMBEDDINGS

To demonstrate how HiM captures hierarchical structure in the learned embeddings, we visualize the hyperbolic representations on a representative semantic hierarchy with WordNet. The hyperbolic embeddings learned by HiM is presented in Figure 2, which illustrates a representative hierarchical path, sport → skating → skateboarding. The HiM-trained embeddings exhibit tight clustering of semantically related nodes (e.g., skating and sport) in hyperbolic space, indicating enhanced semantic alignment. Moreover, the embeddings clearly capture the hierarchical structure, as higher-level concepts such as sport are positioned closer to the origin, while more specific

concepts like `skateboarding` are embedded farther from the origin in a compact and organized manner. More details and geometric analysis of these hyperbolic embeddings, including quantitative metrics comparing h-norms, geodesic distances, and hierarchical depth correlations across both Poincaré and Lorentz manifolds, is provided in Appendix A.5.

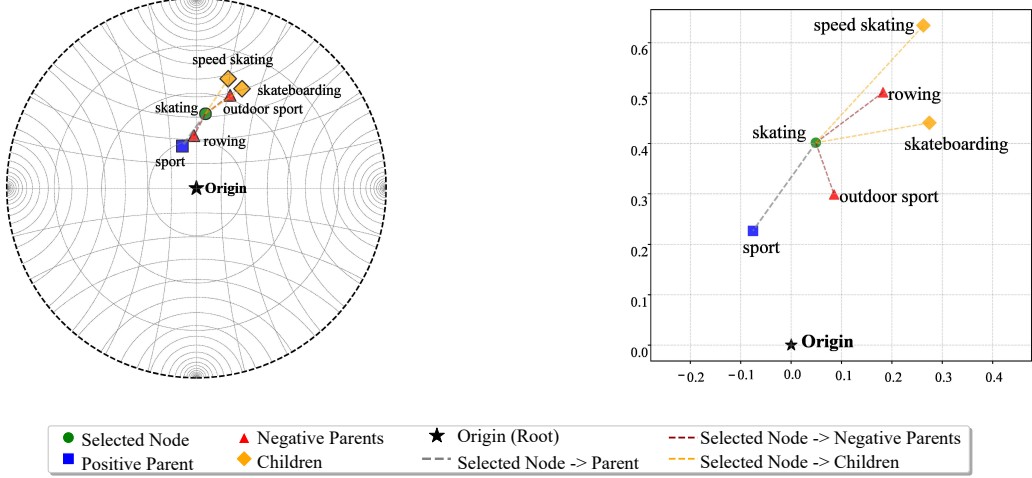

Figure 2: Visualization of HiM's embeddings trained on the WordNet dataset in the Poincaré ball manifold. **Left:** The full hyperbolic space, illustrating the distribution of entities with parent nodes positioned closer to the origin and child nodes extending toward the boundary, reflecting the exponential expansion of hyperbolic geometry. **Right:** A zoomed-in view emphasizing the hierarchical structure, such as the path sport → skating → skateboarding. Dots represent the entities, with colors indicating hierarchical relationships. For a selected node "skating" denoted by the green dot, the blue node denotes its parent nodes (e.g., sport), and red indicates its hard negatives, such as siblings/cousins (e.g., rowing). Yellow nodes (e.g, skateboarding, speed skating) indicate children nodes of the selected node (skating), meaning the grandchildren nodes of the blue node (sport).

## 6 CONCLUSION

By integrating hyperbolic embeddings in the model, HiM successfully captures hierarchical relationships in complex long-range datasets, providing a scalable and effective approach for handling long-range dependencies. HiM's unique approach, especially in hyperbolic embedding and its SSM incorporation, showcases its strengths in hierarchical long-range classification, marking it as a significant advancement in hierarchical learning models. Additionally, we find HiM to be more robust in training, primarily due to the Mamba2 blocks' efficient memory usage and the synergy between hyperbolic geometry and SSM-based sequence modeling.

Lorentz embeddings can provide a more natural fit for large-scale datasets with intricate hierarchical patterns compared to other geometries, potentially enhancing performance and interpretability. By demonstrating how a Lorentzian manifold can be effectively deployed for hyperbolic sentence representations, this paper aims to motivate further exploration of hyperbolic geometry in diverse real-world applications, ultimately broadening the scope and impact of geometry-aware neural architectures. Investigating HiM's potential for efficient temporal dependency modeling in intricate long-range hierarchical classification tasks holds significant promise and study its practical applications. Future work could explore integrating CLIP-style pretraining to incorporate multimodal data (e.g., text and images) for tasks like visual question answering, or potentially building on work such as Cobra (Zhao et al., 2025), which demonstrates the potential of extending Mamba models for efficient multimodal language modeling.

## ETHICS STATEMENT

This work focuses on developing computational methods for hierarchical language understanding and does not involve human subjects or sensitive data collection. The datasets used (DOID, FoodOn, WordNet, SNOMED-CT) are publicly available ontologies that do not contain personally identifiable information.

## REPRODUCIBILITY STATEMENT

To ensure reproducibility of our results, we provide comprehensive implementation details in Section 4 including hyperparameters, training procedures, and evaluation metrics. The source code for HiM, including model architecture, hyperbolic projection operations, and loss functions, is publicly available at `https://github.com/BerryByte/HiM`.

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

## A  APPENDIX

### A.1  PRELIMINARIES

#### A.1.1  HYPERBOLIC GEOMETRY

In hyperbolic geometry, the notion of curvature is commonly represented by negative curvature $\mathcal{K} = -\frac{1}{c}$, where $c > 0$. Equivalently, one may define a 'radius' $r = \sqrt{c}$. A smaller radius $r$ corresponds to a larger and higher negative curvature ($\mathcal{K}$), effectively making the hyperbolic manifold more curved, granting more flexibility to the hierarchical depth. Conversely, letting $r \to \infty$ approaches flat (Euclidean) space since $\mathcal{K} \to 0$. Basically, $r$ controls the rate of exponential expansion on the hyperbolic manifold. This choice impacts how data at varying levels of abstraction distributes on the manifold and is crucial for tasks requiring fine-grained or exponential separation of hierarchical data. By leveraging hyperbolic space, language models can encode features more naturally in a hierarchical branching, keeping more generalized features located near the root of the hierarchy tree, i.e., near the origin of the Hyperbolic Manifold, and the more specific or complex entities are branched further from the origin towards the margin.

A popular way to realize hyperbolic geometry in an $n$-dimensional setting is via the Poincaré ball model. Here, the underlying space is the open Poincaré unit ball:

$$\mathcal{B}^n = \{\mathbf{x} \in \mathbb{R}^n : \|\mathbf{x}\| < \sqrt{c}\}, \tag{10}$$

equipped with a metric tensor that expands distances near the boundary. Concretely, each point $\mathbf{x}$ in the ball maintains a local geometry that grows increasingly "stretched" as $\|\mathbf{x}\|$ approaches radius $\sqrt{c}$. Formally, the distance between two points $\mathbf{x}$ and $\mathbf{y}$ in a Poincaré ball is computed by

$$d_{\mathcal{P}}(\mathbf{x}, \mathbf{y}) = \sqrt{c} \cdot \text{arcosh}\left(1 + 2 \frac{\|\mathbf{x} - \mathbf{y}\|^2}{(1 - \|\mathbf{x}\|^2/c)(1 - \|\mathbf{y}\|^2/c)}\right). \tag{11}$$

This representation has gained attention in machine learning due to relatively simple re-parameterizations for gradient-based updates, thus facilitating the embedding of hierarchically structured data (Nickel & Kiela, 2017).

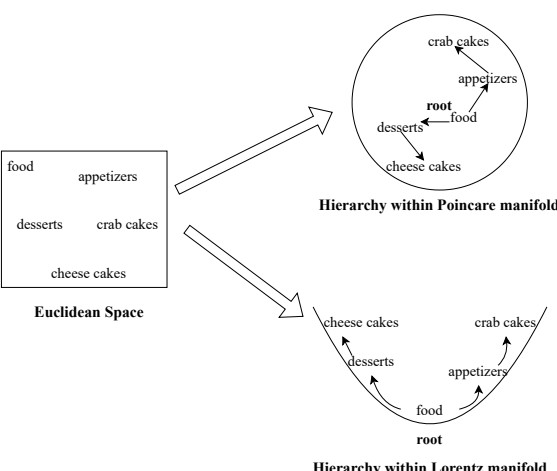

Figure 3: Illustration of word embeddings in Euclidean (Left) vs. Hyperbolic Spaces for hierarchical representation in Poincaré (Top right) and Lorentzian Manifolds (Bottom right).

While the Poincaré ball confines all points within the unit sphere (Equation 10), the Lorentzian manifold leverages an $(n + 1)$-dimensional Minkowski space (Equation 12), enabling a different perspective on hyperbolic geometry. Specifically, points reside on the "hyperboloid" defined by:

$$\mathcal{L}^n = \left\{ \mathbf{x} \in \mathbb{R}^{n+1} : \langle \mathbf{x}, \mathbf{x} \rangle_M = -c, \ x_0 > 0 \right\}, \tag{12}$$

where $\langle \cdot, \cdot \rangle_M$ denotes the Minkowski inner product, typically $-x_0 y_0 + \sum_{i=1}^n x_i y_i$. The hyperbolic distance between two points $\mathbf{x}$ and $\mathbf{y}$ then appears in the form:

$$d_{\mathcal{L}}(\mathbf{x}, \mathbf{y}) = \sqrt{c} \cdot \text{arcosh} \left( -\frac{\langle \mathbf{x}, \mathbf{y} \rangle_M}{c} \right). \tag{13}$$

Compared to the Poincaré ball, this approach can sidestep certain numerical instabilities near the boundary because vectors are not constrained to lie within a finite radius. Moreover, Lorentz-based formulations often allow more direct computation of geodesics and exponential maps, making them advantageous for large-scale hyperbolic embeddings (Nickel & Kiela, 2018). Krioukov et al. (2010) provides a theoretical foundation for the hyperbolic geometry of complex networks, showing that many real-world networks naturally embed into hyperbolic spaces, supporting our choice of the Poincaré and Lorentzian models for hierarchical language embeddings.

We can see how hierarchies are represented differently in Euclidean space, the Poincaré manifold, and the Lorentzian manifold as illustrated in Figure 3. On the left, the hierarchical structure is arranged as a standard tree. While the relationships are maintained, Euclidean space does not naturally encode hierarchical distances in 2D. In Figure 3 the upper-right diagram shows the hierarchy embedded into the Poincaré ball (the root/origin being at the center). The more generalized parent nodes are positioned near origin, and descendant nodes extend outward near the margin. This representation captures the exponential growth of hierarchical structures, where sibling nodes are placed far apart in terms of geodesic distance. The lower-right diagram visualizes the same hierarchy embedded in the Lorentz hyperboloid. The Lorentzian manifold in $\mathbb{R}^{n+1}$ consists of $n$ spatial dimensions and one time-like dimension ($x_0$). The origin is at the center-bottom of the Hyperboloid, and nodes are arranged along the hyperboloid surface. More generalized parent nodes are positioned near the bottom, and the descendants keep extending upward on the cone of Lorentz. Unlike the Poincaré model, which confines embeddings within a finite ball, the Lorentz model represents hierarchies in an unbounded space, making it particularly suitable for representing deeply nested hierarchies.

### A.1.2 MAMBA2

Mamba2 is a state-space model (SSM) introduced by Tri Dao and Albert Gu that refines the original Mamba architecture with improved performance and simplified design (Dao & Gu, 2024). Mamba-2

builds upon the original Mamba architecture by introducing the State Space Duality (SSD) framework, which establishes theoretical connections between State Space Models (SSMs) and attention mechanisms. Mamba2 achieves 2-8× faster processing while maintaining competitive performance compared to Transformers for language modeling tasks. In order to formulate the overall computation for a single Mamba2 block, let $\mathbf{x}_{1:L} = [\mathbf{x}_1, \ldots, \mathbf{x}_L]$ be the token (or embedding) sequence for a given input. A single Mamba2 block transforms $\mathbf{x}_{1:L}$ into an output sequence $\mathbf{y}_{1:L}$. Each input token embedding $\mathbf{x}_t \in \mathbb{R}^d$ is first normalized via RMSNorm. RMS normalization ensures that the norm of the embeddings remains stable across different inputs, preventing extreme values from causing instability during training.

$$\widetilde{\mathbf{x}}_t = \text{RMSNorm}(\mathbf{x}_t). \tag{14}$$

Given weights $W$ and bias $b$, we project the input into a higher-dimensional space to obtain $u$.

$$\mathbf{u}_t = W_{\text{in}}\, \widetilde{\mathbf{x}}_t + \mathbf{b}_{\text{in}}, \tag{15}$$

yielding $\mathbf{u}_t \in \mathbb{R}^{d'}$. $u_t$ is split it into two components, $x'_t$ and $z'_t$.

$$u_t = \begin{bmatrix} \mathbf{x}'_t \\ \mathbf{z}'_t \end{bmatrix}. \tag{16}$$

The component $z'_t$ is reserved for the gating mechanism used later in the process. For each time step $t$, hidden states $\mathbf{h}_t$ evolve under:

$$\mathbf{h}_t = A\, \mathbf{h}_{t-1} + B\, \mathbf{u}_t, \quad \mathbf{z}_t = C\, \mathbf{h}_t, \tag{17}$$

where $A$ is the state transition matrix, $B$ is the input matrix, and $C$ is the output matrix. In our case, these kernels are with dimensions $\mathbf{A} \in \mathbb{R}^{96 \times 96}$, $\mathbf{B} \in \mathbb{R}^{96 \times 768}$, and $\mathbf{C} \in \mathbb{R}^{768 \times 96}$. To enable efficient $O(L)$ complexity, Mamba2 uses *structured* versions of $A, B, C$ (e.g., diagonal-plus-low-rank forms) and fast transforms (such as FFT-based convolution). Mamba2 incorporates a gating mechanism to blend the output of the state-space layer back with the original input, thus forming a residual block:

$$\mathbf{y}_t = \sigma(\mathbf{g}_t)\, \mathbf{z}_t + \mathbf{x}_t, \quad \mathbf{g}_t = W_g\, \widetilde{\mathbf{x}}_t + \mathbf{b}_g, \tag{18}$$

where $\sigma(\cdot)$ is typically a SiLU activation that follows this operation for non-linearity.

$$\sigma(x) = x \cdot sigmoid(x), \quad sigmoid(x) = \frac{1}{1 + e^{-x}}. \tag{19}$$

This gating helps regulate the flow of information and provides additional stability during training.

Mamba2 establishes a theoretical framework connecting SSMs and attention mechanisms through "state space duality", allowing the model to function either as an SSM or as a structured form of attention via the below formulation:

$$L := 1\text{SS}(a) \quad \text{and} \quad M = L \circ \left( C\, B^\top \right). \tag{20}$$

where $L$ is the semiseparable matrix structure derived from the state matrix A, $\circ$ denotes the hadamard product.

This paper introduces a Mamba2-based LLM known as SentenceMamba-16M, a lightweight model with 16M parameters, suitable for resource-efficient and high-quality sentence embedding generation. As we can see in Figure 1, we incorporate 4 Mamba2 blocks in our SentenceMamba-16M for efficient state-space modeling.

## A.2 HYPERBOLIC LOSS CALCULATIONS

Depending on our dataset, we can either apply Triplet Loss when we have a triplet relationship in our data and need to enforce relative distance constraints or apply Contrastive Loss when we have pairwise relationships in our data and need to classify pairs as similar or dissimilar. Based on either triplet relationship data or pairwise relationship data, we perform triplet loss or contrastive loss calculation. Then, we calculate the weighted loss of centripetal loss and clustering loss as the hyperbolic loss. Hyperbolic loss computation framework is illustrated in Figure 4, which shows how these two loss components are weighted and combined to create the final training objective. Loss minimization involves clustering related entities and distancing unrelated ones (clustering loss) and tightening parent entities closer to the hyperbolic manifold's origin than their child counterparts (centripetal loss), giving it a hierarchical structure.

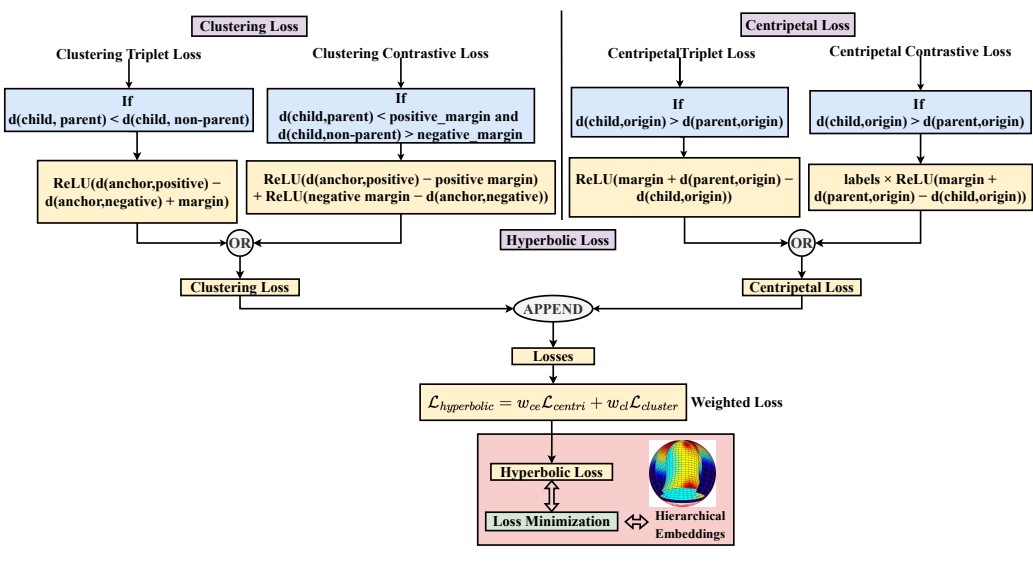

Figure 4: Calculation of hyperbolic loss from clustering loss and centripetal loss.

## A.3  DATASET STATISTICS

To provide a comprehensive overview of the datasets used in our experiments, we detail the size, number of entities (nodes), and train/validation/test splits for each dataset in Table 3. These datasets are represented as directed acyclic graphs (DAGs), where nodes denote entities (e.g., diseases in DOID, synsets in WordNet) and edges denote direct subsumption relations (is-a). Splits are created by sampling direct ($E$) and indirect (multi-hop, $T$) subsumptions, ensuring coverage of both mixed-hop prediction and multi-hop inference tasks.

Table 3: Statistics of hierarchical ontology datasets

| Dataset | #Entities | #DirectSub | #IndirectSub | Splits (Train/Val/Test) |
|---|---|---|---|---|
| **DOID** | 11,157 | 11,180 | 45,383 | Mixed-hop: 111K / 31K / 31K |
| **FoodOn** | 30,963 | 36,486 | 438,266 | Mixed-hop: 361K / 261K / 261K |
| **WordNet (Noun)** | 74,401 | 75,850 | 587,658 | Multi-hop: 834K / 323K / 323K |
| | | | | Mixed-hop: 751K / 365K / 365K |
| **SNOMED-CT** | 364,352 | 420,193 | 2,775,696 | Multi-hop: 4,160K /1,758K/1,758K |
| | | | | Mixed-hop: 4,160K/1,758K/1,758K |

## A.4  TASK FORMULATIONS

### A.4.1  MULTI-HOP INFERENCE

Let $G = (V, E)$ denote a hierarchical graph, where $V$ represents entities (nodes) and $E$ denotes direct subsumption edges (e.g., parent-child relationships). The transitive closure $T$ of $E$ encompasses all indirect (multi-hop) subsumptions, such as relationships spanning two or more hops (e.g., grandparent-to-grandchild). The multi-hop inference task trains a model $f_{\text{MI}}$ on the direct edges $E$ and evaluates its ability to predict the existence of unseen indirect relations in $T$:

$$f_{\text{MI}} : (V, E) \rightarrow \hat{T}, \tag{21}$$

where $\hat{T}$ approximates $T$. This binary classification task tests transitive reasoning, such as inferring "dog is a vertebrate" from "dog is a mammal" and "mammal is a vertebrate." The model computes hyperbolic distances between entity embeddings, with a threshold determining relationship existence.

Evaluation uses precision, recall, and $F1_{\text{MI}}$ scores over test pairs sampled from $T \cup N$, where $N$ represents negative pairs (non-subsumptions). Test sets $S_{\text{test}}$ are constructed as:

$$S_{\text{test}} = \{(v_i, v_j) \mid (v_i, v_j) \in T \cup N, |N| = 10|T|\}, \tag{22}$$

with negatives, including hard cases like sibling entities (sharing a parent but not directly or transitively linked). This assesses fine-grained discrimination across both upward (child-to-ancestor) and downward (parent-to-descendant) directions, leveraging HiM's hyperbolic embeddings.

### A.4.2 MIXED-HOP PREDICTION

The mixed-hop prediction task evaluates the model's ability to predict the exact number of hops between entities, encompassing both direct (1-hop) and multi-hop (2+ hops) subsumptions. Given a training subset $E$, the model $f_{\text{MP}}$ is trained and tested on:

$$f_{\text{MP}} : (V, E) \to \hat{R}, \tag{23}$$

where $R = E \cup T$ includes all held-out direct and transitive subsumptions, and $\hat{R}$ approximates $R$. Unlike multi-hop inference, which focuses on existence, mixed-hop prediction quantifies hierarchical distance (e.g., 1, 2, or 3 hops), such as distinguishing "dog to mammal" (1 hop) from "dog to vertebrate" (2 hops). This is framed as a multi-class classification task, mapping hyperbolic distances to discrete hop counts.

Evaluation employs $F1_{\text{MP}}$ scores over test sets:

$$S_{\text{test}} = \{(v_i, v_j) \mid (v_i, v_j) \in (E \cup T) \cup N, |N| = 10|E \cup T|\}, \tag{24}$$

where positive pairs from $E \cup T$ are labeled with their true hop distances, and negatives (e.g., siblings or unrelated entities) are included at a 1:10 ratio. Hard negatives, such as sibling pairs sharing a parent $v_k$ without a subsumption link, enhance the task's difficulty. This bidirectional task also assesses reasoning in both upward and downward directions.

### A.5 LEARNING INTERPRETABLE HIERARCHICAL SEMANTICS THROUGH HYPERBOLIC GEOMETRY

To provide more interpretable results of our HiM models for the hierarchical learning, we conducted a deeper geometric analysis of the hyperbolic entity embeddings for semantically related WordNet entities under **HiM-Poincaré** and **HiM-Lorentz** manifolds (see the visualization of hyperbolic embeddings in Figure 2). Specifically, we computed three key metrics with the learned hyperbolic embeddings: 1) "**hyperbolic geodesic distances**" between each pair of entities, 2) "**h-norm**" represents the norm distance from the origin, a higher h-norm often indicates a deeper or more specific concept in the hierarchy, 3) "**depth**" is the WordNet tree depth. In both sets of entities, the h-norm correlates strongly with the hierarchical depth, see Figure 5. For instance, in Table 4, the entity `sport` (depth 9) has an h-norm of 0.55, while `skateboarding` (depth 12) has an h-norm of 2.25, reflecting the hierarchical expansion from general to specific concepts. However, a key difference emerges when comparing the two manifolds: **HiM-Lorentz** consistently produces smaller hyperbolic distances and h-norms compared to **HiM-Poincaré**. For example, in Table 5 (HiM-Lorentz), the parent-child relationships (e.g., *sport $\to$ skating $\to$ skateboarding*) have tighter distances and h-norm gradients compared to Table 4.

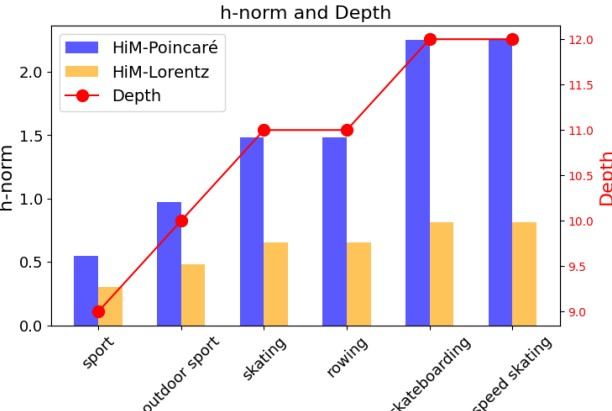

Figure 5: Alignment between the computed h-norms (derived from hyperbolic embeddings by HiM-Poinaré and HiM-Lorentz) and the actual tree-depth for sports-related entities in the WordNet dataset. As the depth increases from general terms like "sport" to specific ones like "skateboarding" and "speed skating", both HiM models show increasing h-norms, reflecting the underlying hierarchical structure. While *HiM-Poincaré produces higher h-norms that better differentiate fine-grained semantic levels*, while *HiM-Lorentz yields more compact yet hierarchy-preserving embeddings with improved numerical stability*. Our results illustrate that both HiM models effectively encode semantic hierarchy, with **Poincaré favoring detail and Lorentz emphasizing robustness**.

This reduction in distances under the Lorentz manifold is advantageous for hierarchical modeling. The Lorentz model's unbounded nature avoids the boundary constraints of the Poincaré ball, which can lead to numerical instability near the boundary. By mapping embeddings into an unbounded hyperboloid, **HiM-Lorentz** achieves tighter clustering of related entities (e.g., between `sport` and `skating`: 0.73 h-Norm of HiM-Lorentz vs. 1.67 h-norm of HiM-Poincaré) while maintaining the hierarchical structure. This tighter clustering enhances the model's ability to distinguish fine-grained relationships, especially in deeper hierarchies, as evidenced by the smaller standard deviations of **HiM-Lorentz** in performance metrics (Table 1).

Table 4: Hyperbolic distances, h-norms, and depths for sports-related entities (from Figure 2) using **HiM-Poincaré**, sorted by increasing depth.

|  | sport | outdoor sport | skating | rowing | skateboarding | speed skating |
|---|---|---|---|---|---|---|
| sport | 0.00 | 1.17 | 1.67 | 1.62 | 2.43 | 2.40 |
| outdoor sport | 1.17 | 0.00 | 1.90 | 1.95 | 2.66 | 2.62 |
| skating | 1.67 | 1.90 | 0.00 | 2.36 | 3.12 | 3.07 |
| rowing | 1.62 | 1.95 | 2.36 | 0.00 | 3.10 | 3.11 |
| skateboarding | 2.43 | 2.66 | 3.12 | 3.10 | 0.00 | 3.76 |
| speed skating | 2.40 | 2.62 | 3.07 | 3.11 | 3.76 | 0.00 |
| **h-norm** | 0.55 | 0.97 | 1.48 | 1.48 | 2.25 | 2.25 |
| **depth** | 9 | 10 | 11 | 11 | 12 | 12 |

A.6    Performance Comparisons between Hyperbolic embeddings and Euclidean embeddings

For `mixed-hop prediction` (Figure 6), **HiM-Lorentz** achieves better performance on datasets with deeper hierarchies, such as DOID ($\delta$-hyperbolicity = 0.019) and SNOMED-CT ($\delta$-hyperbolicity = 0.026). This aligns with the Lorentz manifold's ability to handle deeply nested structures more effectively. However, for FoodOn ($\delta$-hyperbolicity = 0.185), **HiM-Poincaré** slightly outperforms **HiM-Lorentz**. FoodOn's higher $\delta$-hyperbolicity indicates a less tree-like structure, suggesting that

Table 5: Hyperbolic distances, h-norms, and depths for sports-related entities (from Figure 2) using **HiM-Lorentz**, sorted by increasing depth.

|  | sport | outdoor sport | skating | rowing | skateboarding | speed skating |
|---|---|---|---|---|---|---|
| sport | 0.00 | 0.55 | 0.73 | 0.71 | 0.87 | 0.87 |
| outdoor sport | 0.55 | 0.00 | 0.79 | 0.87 | 0.98 | 0.95 |
| skating | 0.73 | 0.79 | 0.00 | 0.95 | 1.08 | 1.08 |
| rowing | 0.71 | 0.87 | 0.95 | 0.00 | 1.07 | 1.07 |
| skateboarding | 0.87 | 0.98 | 1.08 | 1.07 | 0.00 | 1.20 |
| speed skating | 0.87 | 0.95 | 1.08 | 1.07 | 1.20 | 0.00 |
| **h-norm** | 0.30 | 0.48 | 0.65 | 0.65 | 0.81 | 0.81 |
| **depth** | 9 | 10 | 11 | 11 | 12 | 12 |

the Poincaré model's bounded nature may better capture less hierarchical relationships in certain contexts. Both hyperbolic models significantly outperform the Euclidean baselines.

In `multi-hop inference` (Figure 7), **HiM-Lorentz** again demonstrates robust performance, particularly on SNOMED-CT and WordNet, which exhibit deeper hierarchies (SNOMED-CT $\delta$-hyperbolicity = 0.0254, WordNet $\delta$-hyperbolicity = 0.1431). The smaller standard deviations in **HiM-Lorentz**'s metrics (e.g., 0.003 for SNOMED-CT F1) compared to **HiM-Poincaré** (0.028) highlight its stability, a benefit of the Lorentz manifold's numerical advantages. Notably, **HiM-Poincaré** achieves a slightly higher recall on SNOMED-CT, suggesting that the bounded nature of the Poincaré ball can occasionally enhance sensitivity. However, the overall F1 score favors **HiM-Lorentz**, indicating better balance in precision and recall.

A key observation across both tasks is the impact of dataset's tree-like structure as measured by $\delta$-hyperbolicity. Datasets with lower $\delta$-hyperbolicity (meaning more tree-like) benefit more from **HiM-Lorentz**, as its unbounded manifold better captures the exponential expansion of deep hierarchies. In contrast, FoodOn's higher $\delta$-hyperbolicity correlates with **HiM-Poincaré**'s better performance, suggesting that the choice of manifold may depend on the dataset's structural properties.

Figures 8 to 10 illustrate the training dynamics of HiM-Poincaré and HiM-Lorentz across epochs for each dataset and task, plotting Hyperbolic Loss and F1 Score. The Hyperbolic Loss decreases steadily for both models across all datasets, indicating effective optimization of hierarchical relationships. HiM-Lorentz often exhibits a slightly faster convergence rate and lower final loss compared to HiM-Poincaré, reflecting the Lorentz manifold's suitability for capturing exponential hierarchical expansion. The F1 Score trends mirror the loss behavior, with HiM-Lorentz often achieving slightly better F1 scores.

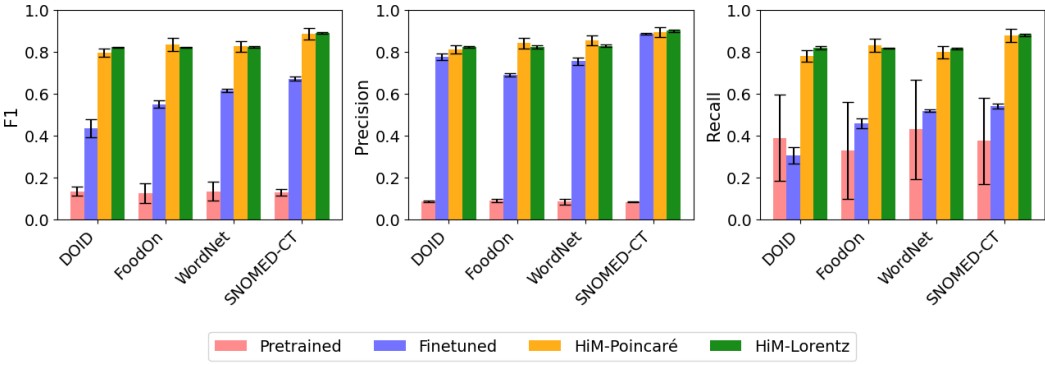

Figure 6: Comparisons of `mixed-hop prediction` performance for DOID, FoodOn, WordNet, and SNOMED-CT datasets based on our proposed hyperbolic mamba (HiM) models–*HiM-Poincaré* and *HiM-Lorentz*, and their Euclidean counterparts–*Pretrained* and *fine-tuned sentenceMamba-16M*.

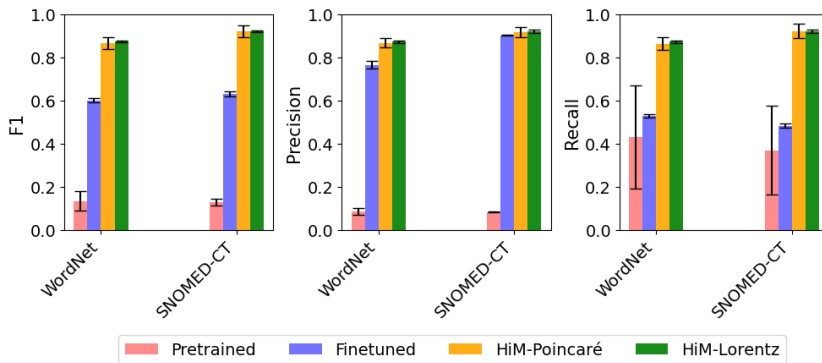

Figure 7: Comparisons of `multi-hop inference` performance for DOID, FoodOn, WordNet, and SNOMED-CT datasets based on our proposed hyperbolic mamba (HiM) models–*HiM-Poincaré* and *HiM-Lorentz*, and their Euclidean counterparts–*Pretrained* and *fine-tuned sentenceMamba-16M*.

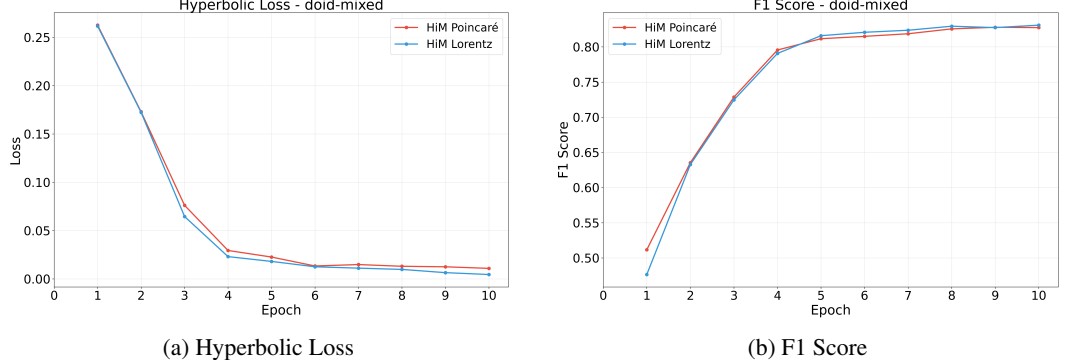

Figure 8: Comparison of hyperbolic loss and F1 score on **DOID mixed-hop prediction** across epochs.

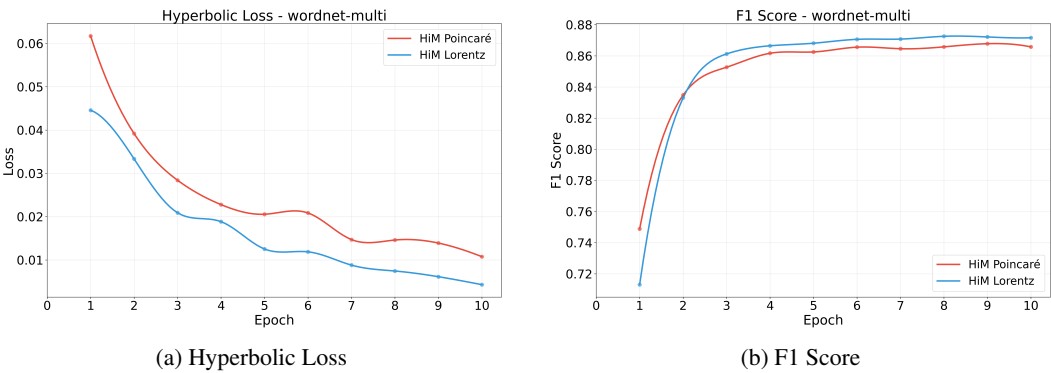

Figure 9: Comparison of hyperbolic loss and F1 score on **WordNet multi-hop inference** across epochs.

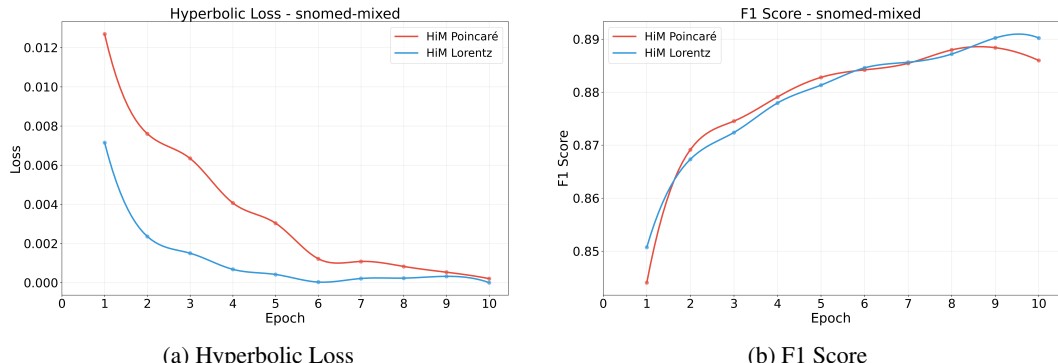

(a) Hyperbolic Loss                    (b) F1 Score

Figure 10: Comparison of hyperbolic loss and F1 score on **SNOMED-CT mixed-hop prediction** across the epochs.

### A.7    ADDITIONAL EXPERIMENTAL ANALYSIS

#### A.7.1    PROMPTED LLM EXPERIMENT (HIM VS. GPT-4O)

To evaluate HiM's performance relative to contemporary large language models, we conducted a zero-shot evaluation using GPT-4o[3] on the WordNet mixed-hop prediction task. This comparison provides insight into how our specialized hyperbolic architecture performs against general-purpose language models that rely on vast pretraining but lack explicit hierarchical inductive biases. We generated 500 binary classification questions following the structure "Is [entity1] a subtype/subclass of [entity2]?" sampled from the same test set used for HiM evaluation. The experimental setup mirrored HiM's training regime: for each sampled child node, we generated one positive question and ten negative questions (corresponding to HiM's 1 positive parent + 10 hard negatives). GPT-4o was provided with a list of 74,401 WordNet entities as context and answered all 500 questions in a single zero-shot prompt without additional training. The results in Table 6 demonstrate that both HiM variants substantially outperform GPT-4o. GPT-4o performs well on general knowledge hierarchies where concepts like 'dog' → 'animal' are well-represented in large-scale training corpora. HiM (both Poincaré and Lorentz) still outperform GPT-4o by a clear margin, even on such general-knowledge hierarchy. The superior performance of HiM highlights the effectiveness of our hyperbolic modeling approach for hierarchical reasoning, even when compared to pretrained LLM with significantly larger parameter counts and extensive pretraining.

Table 6: F1 scores comparing HiM models with GPT-4o on WordNet mixed-hop prediction task

| Dataset | HiM | | GPT-4o |
|---|---|---|---|
| | Poincaré | Lorentz | |
| WordNet-mixed | 0.859 | 0.850 | 0.750 |

#### A.7.2    COMPUTATIONAL EFFICIENCY ANALYSIS

To substantiate our claims regarding Mamba's linear complexity advantages, we conducted a sequence length scaling study on the WordNet dataset for the mixed-hop prediction task. The results in Table 7 validate Mamba's theoretical linear complexity characteristics. Doubling the sequence length from 128 to 256 tokens results in an exact 2× increase in both FLOPs (4.46G → 8.9G) and MACs (2.23G → 4.45G), while memory consumption remains constant at 66.91MB due to the fixed number of model parameters and activations. This linear scaling behavior contrasts sharply with transformer-based architectures, where sequence length increases would result in quadratic growth in computational requirements.

---

[3]https://openai.com/index/hello-gpt-4o/

Table 7: Sequence length scaling analysis for HiM-16M-Poincaré on WordNet mixed-hop prediction

| Sequence Length | FLOPs | MACs | Memory | Training Time |
|---|---|---|---|---|
| 128 | 4.46 G | 2.23 G | 66.91 MB | 0:41:32 |
| 256 | 8.90 G | 4.45 G | 66.91 MB | 0:42:13 |

## A.8 CODE AVAILABILITY

The source code for the Hierarchical Mamba (HiM) model is publicly available at `https://github.com/BerryByte/HiM` with detailed instructions for setup and execution.

## A.9 DECLARATION OF LLM USAGE

LLMs were only used to assist with writing and formatting, not as part of the core methodology. Large Language Models (LLMs) were used in a limited capacity during the preparation of this manuscript for grammar checking and text refinement. All technical contributions, results and insights are the original work of the authors.

