# OpenReview forum: "Hierarchical Mamba Meets Hyperbolic Geometry: A New Paradigm for Structured Language Embeddings"
_ICLR.cc/2026/Conference — ICLR 2026 Conference Withdrawn Submission_

### Official Review · Reviewer_gJLi · 2025-10-29

**Soundness:** 2
**Presentation:** 2
**Contribution:** 2
**Rating:** 2
**Confidence:** 4

**Summary:**

The paper proposes Hierarchical Mamba (HiM), a model that integrates the Mamba2 state-space model with hyperbolic geometry (both Poincaré and Lorentz manifolds) for hierarchical reasoning tasks such as ontology and taxonomy inference. It introduces a SentenceMamba-16M encoder for sequence embedding and projects outputs into hyperbolic space via learnable curvature. A hyperbolic loss combining centripetal and clustering terms enforces parent–child relationships.
Experiments on WordNet, DOID, FoodOn, and SNOMED-CT show improved F1 scores over Euclidean and hyperbolic transformer baselines, with claims of better scalability for long sequences.

**Strengths:**

1) First work of intergrating Mamba2 and hyperbolic embeddings.

2) The paper includes both Poincaré and Lorentz formulations, curvature regularization, and stability approximations.

3) empirical study across four ontology datasets, including δ-hyperbolicity analysis and visualization of embedding hierarchies.

**Weaknesses:**

1) Questionable motivation for long-range dependency modeling. Ontology reasoning is structurally hierarchical but not sequential. It’s unclear why Mamba’s sequence modeling is advantageous here. The link between “long-range dependencies” and “multi-hop ontology reasoning” is weak and mismatched.

2) Limited novelty. Integrating a known efficient sequence model (Mamba2) with hyperbolic embeddings is incremental, given prior works like SHMamba (2024) and Hyperbolic BERT / HiT (He et al 2024) / HypLoRA (Yang et al) already address hierarchy in both Euclidean and hyperbolic spaces.

3) The “hyperbolic loss” largely reuses existing centripetal and triplet loss ideas without strong new formulation.

4) The paper mixes terminology between hierarchical, long-range, and multi-hop reasoning without a clear theoretical bridge.

5) Evaluation tasks are limited. Only ontology/knowledge graphs datasets are tested. There is no evidence of generalization to natural language or other structured tasks.

**Questions:**

1) Why does ontology reasoning require long-range sequence modeling? Are there sequential dependencies that justify using Mamba rather than a hyperbolic Transformer like HiT?

2) How is HiM different from SHMamba (2024), HiT (2024), or Hyperbolic LoRA (2024) beyond substituting the backbone?

3) Can you quantify how much of the performance gain arises from Mamba2 vs. the hyperbolic embedding itself?

4) Have you compared HiM with strong non-hyperbolic ontology reasoning baselines like KG embeddings?

5) What does “long-range dependency” mean in hierarchical reasoning? Can you formalize or visualize this relationship?

6) How does the model handle non-tree or cyclic structures present in real ontologies like SNOMED-CT?

7) Does learnable curvature consistently converge, or is it dataset-dependent?

8) Why not test the approach on natural language tasks?

---

### Official Review · Reviewer_A5xX · 2025-10-31

**Soundness:** 2
**Presentation:** 2
**Contribution:** 2
**Rating:** 4
**Confidence:** 2

**Summary:**

This paper introduces Hierarchical Mamba (HiM), a new model architecture designed to efficiently create language embeddings that capture hierarchical relationships. The authors identify two main problems with existing models like Transformers: they use flat Euclidean embeddings, which are poorly suited for tree-like data, and their attention mechanisms are computationally expensive.HiM solves this by combining the Mamba2 state-space model with hyperbolic geometry. It uses Mamba2 as an efficient $O(L)$ encoder to process long sequences, creating a 16-million parameter model called SentenceMamba-16M. The model's output is then mean-pooled and projected into a hyperbolic space (either the Poincaré ball or the Lorentzian manifold), which naturally represents hierarchical structures.The model is trained with novel hyperbolic loss functions (centripetal and clustering) to explicitly organize the embeddings, pulling parent nodes toward the origin and grouping related child nodes. Experiments on four ontology datasets (DOID, FoodOn, WordNet, SNOMED-CT) show that HiM significantly outperforms both Euclidean baselines and a hyperbolic Transformer (HiT). The paper also finds that the HiM-Lorentz variant offers more stable and compact embeddings, particularly for datasets with deep, tree-like structures.

**Strengths:**

The core idea of combining the $O(L)$ efficiency of Mamba2 with the $O(L)$ representational power of hyperbolic geometry for hierarchical data is novel and well-motivated.

The introduction of the SentenceMamba-16M model provides a lightweight and efficient backbone for sentence embedding tasks.

The inclusion of a zero-shot comparison against GPT-4o on the WordNet task is a strong addition, demonstrating that a small, specialized model can outperform a massive, general-purpose one on a specific reasoning task.

**Weaknesses:**

The paper's claim that Mamba2's selective properties are key to its success is not fully proven. Since the methodology applies mean pooling to the Mamba2 block outputs to get a single vector, it is unclear if Mamba's selectivity is providing a benefit beyond just being an efficient $O(L)$ encoder (maybe LSTM?)

A critical detail seems vague. The paper mentions a "geometric stabilization technique that periodically projects the model parameters back onto the manifold" every 100 steps. It does not specify which parameters or the mathematical operation used for this projection, making it difficult to replicate.

The use of both a "learnable curvature" $c$ and a "learnable scaling parameter" $\gamma$ seems redundant. While justified as a stability measure, their interaction (creating an $\mathcal{K}_{eff}$) isn't deeply explored, and it's unclear if this dual-parameter approach is truly necessary.

**NOTE**: The model projects data onto the Lorentz manifold using hyperbolic cosine ($\cosh$) and sine ($\sinh$) functions. Unlike the tanh function used for the Poincaré model, these functions are unbounded and grow exponentially. This creates a high risk of numerical overflow (getting numbers too big for the computer to handle) and exploding gradients, which would cause the training to fail.

Therefore, as I have not independently verified the correct implementation of these highly advanced and sensitive numerical methods, my confidence in a full assessment of this paper's technical soundness is properly lowered.

**Questions:**

Regarding the stabilization technique, could you please provide the exact mathematical formula used to "project the model parameters back onto the manifold"? Which specific parameters (e.g., the entire Mamba2 block, just the final projection layer?) does this operation apply to?

You attribute the model's success to Mamba's selective mechanism. Given the use of mean pooling, did you run any experiments to isolate this? For example, how does HiM's performance compare to a model that replaces Mamba2 with a different $O(L)$ encoder, like an LSTM, but uses the same hyperbolic projection and loss functions?

Why did you choose to implement a dual-parameter system for the geometry (learnable curvature $c$ and learnable scaling $\gamma$) rather than just learning the curvature $c$ alone, which is a more common approach? Did you find that learning $c$ by itself was too unstable?

The terminology for the baselines in Table 1 is confusing. The "Finetuned" model is described as being randomly initialized and trained from scratch, making "Finetuned" a misnomer that could confuse the reader. Could you clarify if I understand this point correctly?

---

### Official Review · Reviewer_B2t6 · 2025-11-01

**Soundness:** 2
**Presentation:** 2
**Contribution:** 2
**Rating:** 4
**Confidence:** 3

**Summary:**

Hierarchical Mamba (HiM) fuses Mamba2 with hyperbolic geometry to learn hierarchy-aware language embeddings, projecting sequences onto the Poincaré ball or Lorentz manifold with learnable curvature and a hyperbolic loss. This design better captures relational distances across hierarchical levels, improving long-range reasoning for mixed-hop and multi-hop inference. Across four linguistic and medical datasets, HiM outperforms Euclidean baselines; the Poincaré variant offers finer-grained distinctions (higher h-norms), while the Lorentz variant yields more stable, compact, hierarchy-preserving embeddings.

**Strengths:**

- Significant performance gains. The method delivers consistently strong improvements across tasks.
- Systematic manifold/curvature study. It compares Poincaré and Lorentz models across multiple datasets and tasks, analyzing how curvature and manifold choice affect performance and stability.

**Weaknesses:**

1. The paper compares only against Mamba, lacking head-to-head evaluations with Hyperbolic Transformers and Euclidean Transformers under matched settings to quantify both accuracy and runtime gains.
2. There is no theoretical or empirical accounting of the constant-factor cost of hyperbolic operations versus Euclidean ones (e.g., training/inference latency,  memory), despite relying on projections/distances (cosh/sinh, exp/log maps).
3. Incomplete approximation analysis. The Maclaurin approximation for cosh/sinh is invoked for |z| < 1e-3, but the trunction order k, error bounds, switching criteria back to exact functions, and fallback for |z| ≥ 1e-3 are unspecified. Please report k, provide an error bound (or empirical error), clarify whether k is fixed or adaptive, and include a small ablation of k vs. training stability/runtime.
4. Overall, the contribution largely applies hyperbolic geometry to Mamba without introducing sufficiently new technical or theoretical innovations.

**Questions:**

Please see Weakness.

---

### Official Review · Reviewer_sr3o · 2025-11-01

**Soundness:** 2
**Presentation:** 2
**Contribution:** 3
**Rating:** 4
**Confidence:** 3

**Summary:**

This paper directly combines Mamba2 SSM with hyperbolic geometry (including the Poincaré ball and Lorentz manifold), utilizing learnable curvature and loss functions tailored for hyperbolic space to generate hierarchy-aware language embeddings.

**Strengths:**

New exploitation, Mamba2+Hyperbolic

**Weaknesses:**

The paper assumes that the fusion of Mamba2 and hyperbolic space can significantly enhance long-sequence and hierarchical reasoning capabilities, but the experiments mainly focus on four preset Ontology-class datasets, failing to cover large-scale real-world scenarios in natural language processing (such as open-domain question answering or multi-document reasoning), resulting in insufficient model generalizability.

Although the authors compared their approach with Euclidean Mamba and Hyperbolic Transformer, there is a lack of direct horizontal evaluation against open-source large models (such as Llama, GPT-4, and stronger Transformer variants), and no comprehensive comparison with state-of-the-art hyperbolic methods or RAG-type methods of similar parameter sizes, which affects the objectivity of the conclusions.

All tasks concentrate on classification for mixed-hop and multi-hop reasoning, neglecting the performance of language embeddings in mainstream tasks such as generation, retrieval, and structured question answering. Moreover, the evaluation metrics are primarily F1, Precision, and Recall, without introducing hierarchy-specific evaluations (such as Tree Edit Distance or hierarchical consistency scores), resulting in insufficient experimental scope.

Regarding the Maclaurin expansions proposed by the authors, this has already been provided in the geoopt repo, which the authors did not cite or refer to.

Regarding these hyperbolic losses, this method is not novel, and for a fair comparison with Euclidean space models, shouldn't you also include such losses to them?

**Questions:**

Although the authors modeled and experimented with both Poincaré and Lorentz isomorphic hyperbolic space manifolds, the paper does not systematically explore whether their combination truly brings irreplaceable expressive power improvement under the premise of theoretical equivalence. Why not consider two Poincaré balls or two Lorentz models? Wouldn't introducing multiple manifolds potentially introduce more instability?

---

### Note · Authors · 2025-11-17

I have read and agree with the venue's withdrawal policy on behalf of myself and my co-authors.